# The effect of genetic variation on promoter usage and enhancer activity

Marco Garieri[1,2,3], Olivier Delaneau[1,2,3], Federico Santoni[1,4], Richard J. Fish[1], David Mull[1], Piero Carninci [5], Emmanouil T. Dermitzakis[1,2,3], Stylianos E. Antonarakis[1,2,4] & Alexandre Fort[1]

The identification of genetic variants affecting gene expression, namely expression quantitative trait loci (eQTLs), has contributed to the understanding of mechanisms underlying human traits and diseases. The majority of these variants map in non-coding regulatory regions of the genome and their identification remains challenging. Here, we use natural genetic variation and CAGE transcriptomes from 154 EBV-transformed lymphoblastoid cell lines, derived from unrelated individuals, to map 5376 and 110 regulatory variants associated with promoter usage (puQTLs) and enhancer activity (eaQTLs), respectively. We characterize five categories of genes associated with puQTLs, distinguishing single from multi-promoter genes. Among multi-promoter genes, we find puQTL effects either specific to a single promoter or to multiple promoters with variable effect orientations. Regulatory variants associated with opposite effects on different mRNA isoforms suggest compensatory mechanisms occurring between alternative promoters. Our analyses identify differential promoter usage and modulation of enhancer activity as molecular mechanisms underlying eQTLs related to regulatory elements.

---

[1] Department of Genetic Medicine and Development, University of Geneva, 1 Michel Servet, Geneva CH1211, Switzerland. [2] Institute of Genetics and Genomics in Geneva, iGe3, 1 Michel Servet, Geneva CH1211, Switzerland. [3] Swiss Institute of Bioinformatics, SIB, UNIL Sorge, Lausanne CH1015, Switzerland. [4] University Hospitals of Geneva, Service of Genetic Medicine, 4 Gabrielle-Perret-Gentil, Geneva CH1205, Switzerland. [5] Division of Genomic Technologies, RIKEN Center for Life Science Technologies, 1-7-22 Suehiro-Cho, Yokohama 230-0045, Japan. Correspondence and requests for materials should be addressed to A.F. (email: alexandre.fort@unige.ch)

For more than a decade, numerous genome-wide association studies (GWAS) have identified thousands of single nucleotide variants (SNVs) associated with human traits and diseases. The contribution of SNVs located within promoter and enhancer elements to disease etiology is well established[1,2]. However, understanding the consequences of these regulatory variants on the human transcriptome remains a major challenge for accurate interpretation of GWAS signals and for the precise identification of causal variants. This issue has been addressed in population studies combining individual genotypes and transcriptome profiles; a design capable of finding associations between SNVs and mRNA levels, namely expression quantitative trait loci (eQTLs)[3,4].

Several observations support the functional implication of eQTLs on gene promoters. First, eQTLs have been recurrently found enriched within promoter regions of their associated genes[4–6]. In addition, regulatory variants have been found associated with alternative transcript usage[5,7,8], including variation in mRNA 5′-end position. Also, given that human genes have, on average, more than four TSSs[9] (transcriptional start sites), differential TSS, or promoter usage deserves to be further investigated to better understand eQTL effects.

Moreover, eQTLs are also enriched in enhancer elements[5,6,10,11]. Briefly, enhancers are *cis*-regulatory regions located remotely from promoters and contribute to the regulation of gene expression by increasing transcription levels and providing information not encoded in proximal promoters, such as the developmental timing or tissue specificity of expression. Enhancers contain binding sites for transcription factors and chromatin looping mediator proteins necessary for them to act on target genes in a distance-independent manner[12]. Yet, the systematic identification of enhancers' target genes requires precise description of enhancer–promoter interactions, based on chromatin conformation assays or functional experiments using genome editing. As a consequence, the target genes of most enhancers remain poorly annotated, increasing the difficulty of interpreting regulatory variants located within enhancers, and in understanding their contribution to human disease.

We hypothesize that mapping genetic variants associated with promoter and enhancer functions can provide novel insights into the mechanism through which eQTLs exert their effects on gene expression. To this end, we quantify genome-wide promoter usage and enhancer activity using CAGE (Cap Analysis of Gene Expression)[13] transcriptome profiling and test the resulting molecular phenotypes for association with nearby genetic variants to discover *cis*-QTLs (Fig. 1a). We report the discovery of 5376 and 110 QTLs that are associated with promoter usage and enhancer activity, respectively. These analyses suggest a strong implication of genetic variants in the molecular regulation of promoter usage. Finally, this study provides an original approach, using CAGE technology, to decipher possible mechanisms of how genetic variation exerts its effect on gene expression through the modulation of enhancer activity.

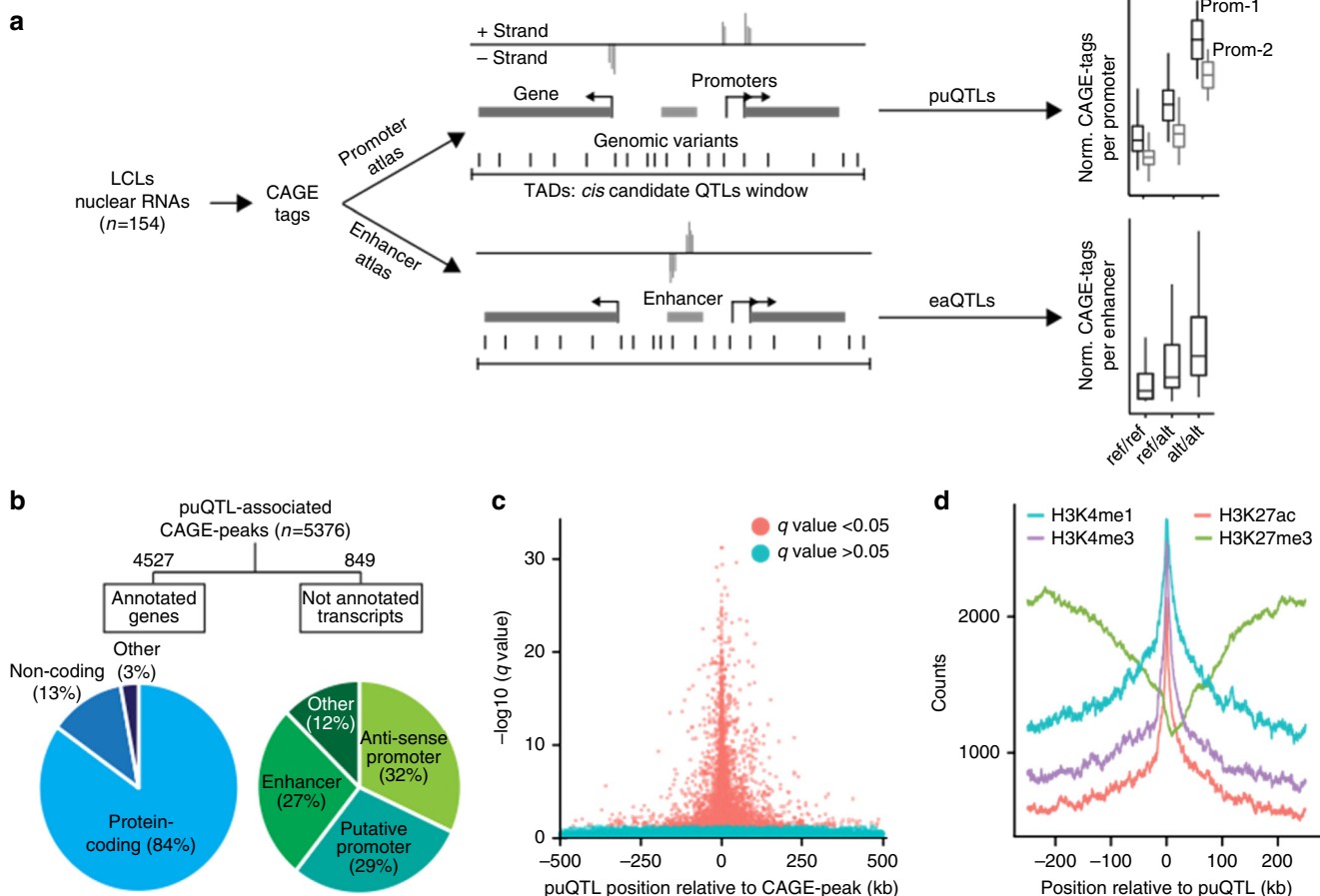

**Fig. 1** Study design and puQTLs mapping. **a** Experimental design for the promoter usage and enhancer activity QTL mapping. **b** Annotation of puQTL-associated CAGE peaks with functional genomic regulatory elements and genomic localization respective to annotated transcripts from GENCODE-V19. **c** Significance relative to distance for each promoter (CAGE peaks) with best-associated puQTL plotted for 1 Mb window. **d** Total counts of histone marks ChIP-seq signals (GM12878 cells, ENCODE data) for the 500 kb regions flanking puQTLs

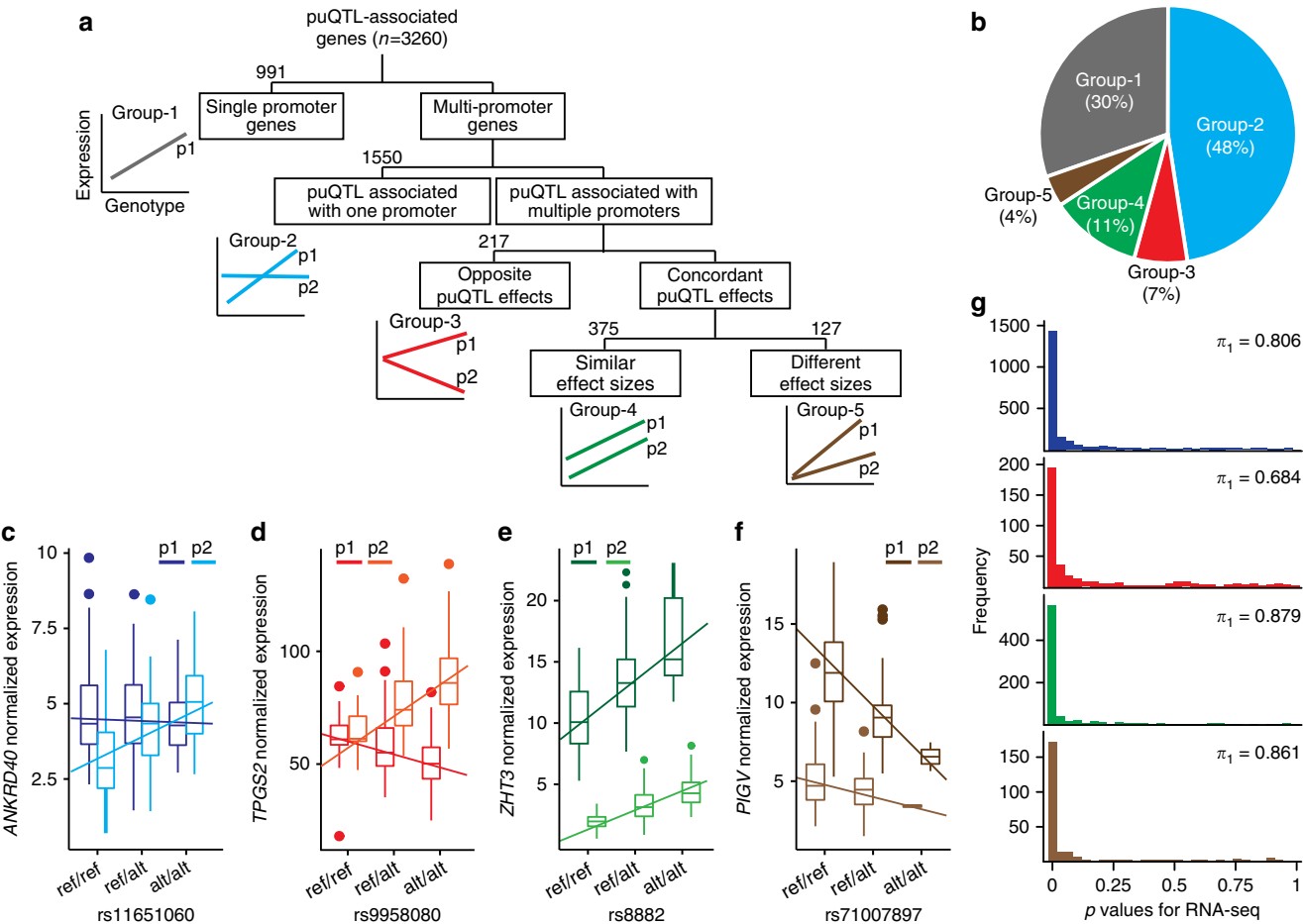

**Fig. 2** puQTL-associated gene classification. **a** Classification procedure for the puQTL-associated genes. **b** Relative size of each group of puQTL-associated genes. **c–f** Representative puQTLs examples for group-2 (**c**), group-3 (**d**), group-4 (**e**), and group-5 (**f**). **g** Replication of puQTLs for each group in RNA-seq data using $\pi_1$ statistics

## Results

**QTL mapping on CAGE transcriptome profiles**. We analyzed 154 transcriptomes from unrelated individuals of central European descent (Supplementary Table 1), applying the CAGE technology to nucleus-enriched total RNA extracted from EBV-transformed lymphoblastoid cell lines (LCLs). CAGE libraries were sequenced at a mean depth of $16.2 \times 10^6$ ($\pm 4.3 \times 10^6$) reads uniquely mapping to the reference genome (Supplementary Fig. 1a). CAGE tags were mapped to the robust CAGE peaks set of the FANTOM promoter atlas[9], yielding, after filtering, 38,759 CAGE peaks across all samples (Supplementary Fig. 1b). According to FANTOM atlas annotations, the CAGE peaks are associated with 13,351 genes and 7424 intergenic TSSs, indicative of potential novel transcripts. Individual genotype information was retrieved from previous studies[14,15] and, following imputation and filtering, a total of 7,508,202 variants were kept for downstream analyses. Any sample mislabeling between sequencing and genotyping data was detected and fixed with an efficient approach[16] that we developed (Supplementary Fig. 1c).

Using normalized CAGE peak expression values and genotypes (Supplementary Fig. 2), we mapped promoter usage-QTLs (puQTLs) in *cis* using the *QTLtools* software[17]. Topologically associating domains[18,19] (TADs, Supplementary Fig. 3a) were used to define the tested *cis*-windows, assuming that proximal and distal regulatory elements acting on a same gene reside within the same TAD. Following this procedure, we mapped 5376 puQTLs at the significance threshold of 5% FDR (Supplementary

Fig. 3b). These puQTLs consist of 4876 unique regulatory variants associated with 5376 CAGE peaks (read promoters) assigned to 2697 protein-coding and 489 non-coding genes, as well as 849 putative novel transcripts. We then combined the human transcript catalog (GENCODE-V19)[20] and histone marks profiling[21] to annotate the puQTL CAGE peaks not associated with genes. The majority of these carry histone marks characteristic of promoter regions ($n = 515$) and can be classified as either antisense promoters ($n = 271$) or putative promoters localized within a gene or in an intergenic region ($n = 244$) (Fig. 1b). Interestingly, 227 CAGE peaks carry histone marks characteristic of enhancer regions and thus can be considered as enhancer-RNAs (eRNAs)[22].

The general features of puQTLs are similar to eQTLs. They localize close to TSSs (Fig. 1c), within open chromatin regions (DNase I hypersensitive sites, Supplementary Fig. 3c) carrying histone modifications specific to active transcription (H3K27ac, H3K4me1, and H3K4me3) and are depleted in regions of repressive H3K27me3 marks (Fig. 1d). We calculated the enrichment of transcription factor-binding sites overlapping puQTLs, using ChIP-seq data (ENCODE data[23], Supplementary Fig. 3d). Among the enriched transcription factors, we detected CEBPB ($p$ value $= 8.99 \times 10^{-5}$) involved in immune and inflammatory responses, IKZF1 ($p$ value $= 9.99 \times 10^{-6}$) implicated in the regulation of lymphocyte differentiation and BCL11A ($p$ value $= 1.09 \times 10^{-4}$) downregulated during hematopoietic cell differentiation and thus possibly associated with B-cell

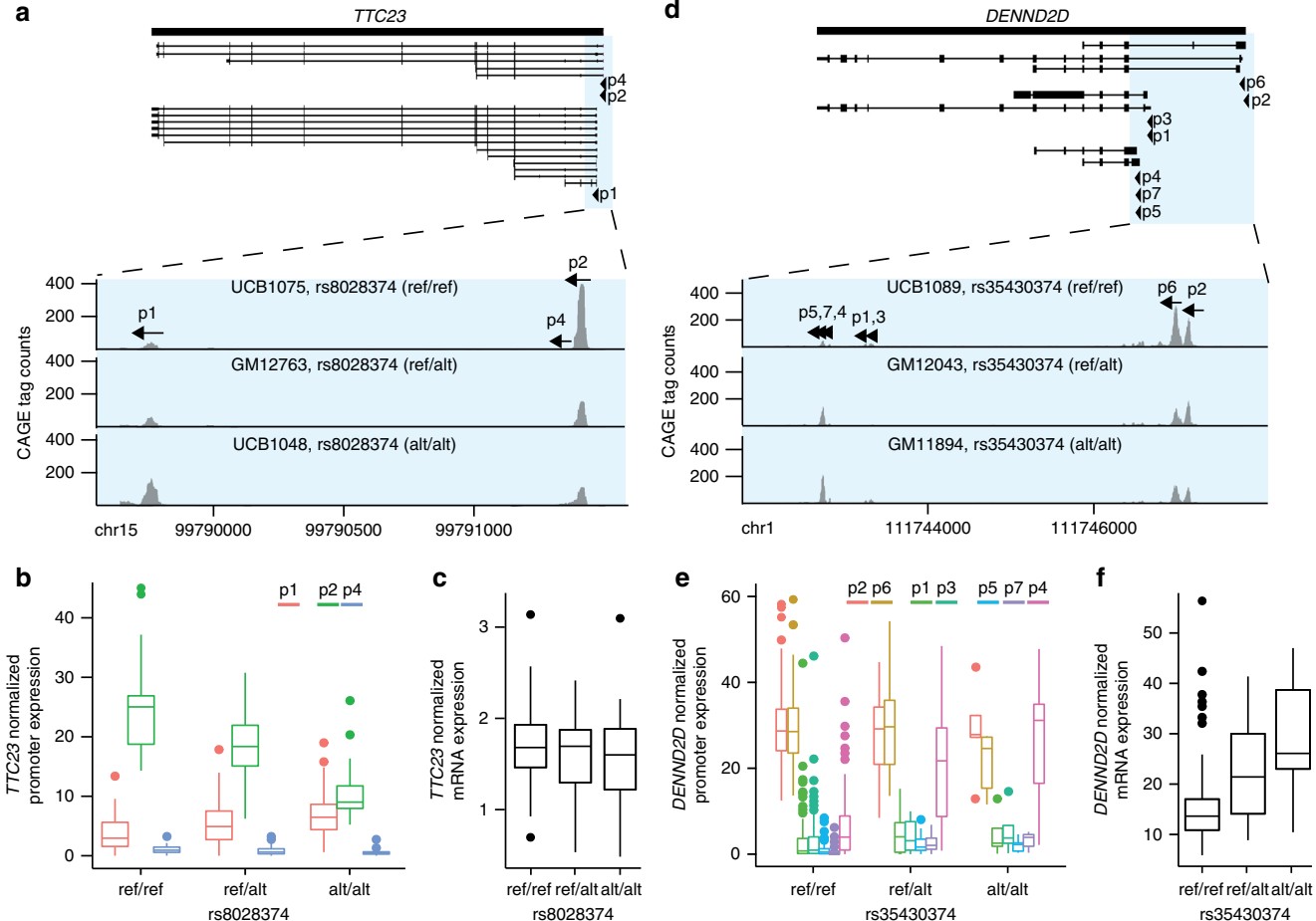

**Fig. 3** Implication of differential promoter usage on mRNA level. Transcript isoforms and row CAGE tag distribution of promoter regions are plotted for individuals of each genotype group for puQTLs associated for the *TTC23* (**a–c**) and *DENND2D* (**d–f**) genes. Normalized promoter (**b**, **e**) and mRNA (**c**, **f**) expression relative to each genotype group are plotted for the entire population (*n* = 154)

malignancies. We also found enrichment for the transcriptional co-activator, EP300 (*p* value = 9.99 × 10$^{-6}$), similarly to the GTEx study[6] observation on splicing-QTLs.

We then estimated the significance of the overlap between puQTLs and disease-associated variants reported in the GWAS catalog[24]. We found 1024 puQTLs (out of 4876 unique variants) overlapping linkage disequilibrium (LD) intervals containing GWAS hits (OR: 1.25, CI: 1.16–1.35, *p* value = 0.0009). Such significant enrichment has been previously reported for QTLs associated with other molecular phenotypes (eQTLs[5,6], splicing QTLs[7], and methylation QTLs[25]). To refine the analysis and decipher whether the effect of the puQTLs and GWAS variants are concordant (i.e., tagging the same functional variant), we applied the regulatory trait concordance (RTC)[26] method. Briefly, RTC accounts for local LD structure and regresses out the genetic effect of the GWAS variant from the CAGE peak expression data, to measure if the puQTLs association is still significant. The RTC scores range between 0 and 1, reflecting not concordant or concordant effects of the pair of puQTL and the GWAS hit, respectively. We found 51 puQTLs passing the high-confidence concordance threshold (RTC > 0.9, Supplementary Table 2) for 35 GWAS hits associated with a variety of phenotypic traits including systemic lupus erythematosus and inflammatory bowel disease. This finding proves the potency of our approach to detect functional variants implicated in disease etiology.

The FANTOM promoter atlas CAGE peaks are ranked, for each gene, according to the total read counts observed in their

data sets[9]. Considering this classification, we find that 52% of the puQTLs are associated with secondary CAGE peaks potentially involving alternative promoters (Supplementary Fig. 4a). In addition, 2289 puQTL-associated genes display more than one CAGE peak and thus likely have several promoters (Supplementary Fig. 4b). Together, these observations suggest that puQTLs are potentially implicated in the regulation of differential promoter usage.

**Regulatory variants associated with promoter usage**. We classified the genes associated with puQTLs into five groups (Fig. 2a, b). Group-1 includes single promoter genes (991 genes) and genes with several CAGE peaks distant by less than 200 nt, which we consider as 5′ RNA variations under the regulation of a single promoter element. Then we considered puQTLs effect sizes (ß, regression slope) and direction to sort multi-promoter genes. First, 1550 multi-promoter genes with puQTLs significantly affecting a single CAGE peak constitute group-2 (Fig. 2c). Group-3 includes 217 multi-promoter genes with puQTLs having opposite effects (ß of different signs) on distant CAGE peaks (Fig. 2d). Group-4 includes 375 multi-promoter genes with puQTLs having concordant effects (ß of same sign) on different CAGE peaks with similar effect sizes (Fig. 2e). Lastly, group-5 includes 127 multi-promoter genes with puQTLs having concordant effects (ß of same sign) on different CAGE peaks with different effect sizes (Fig. 2f).

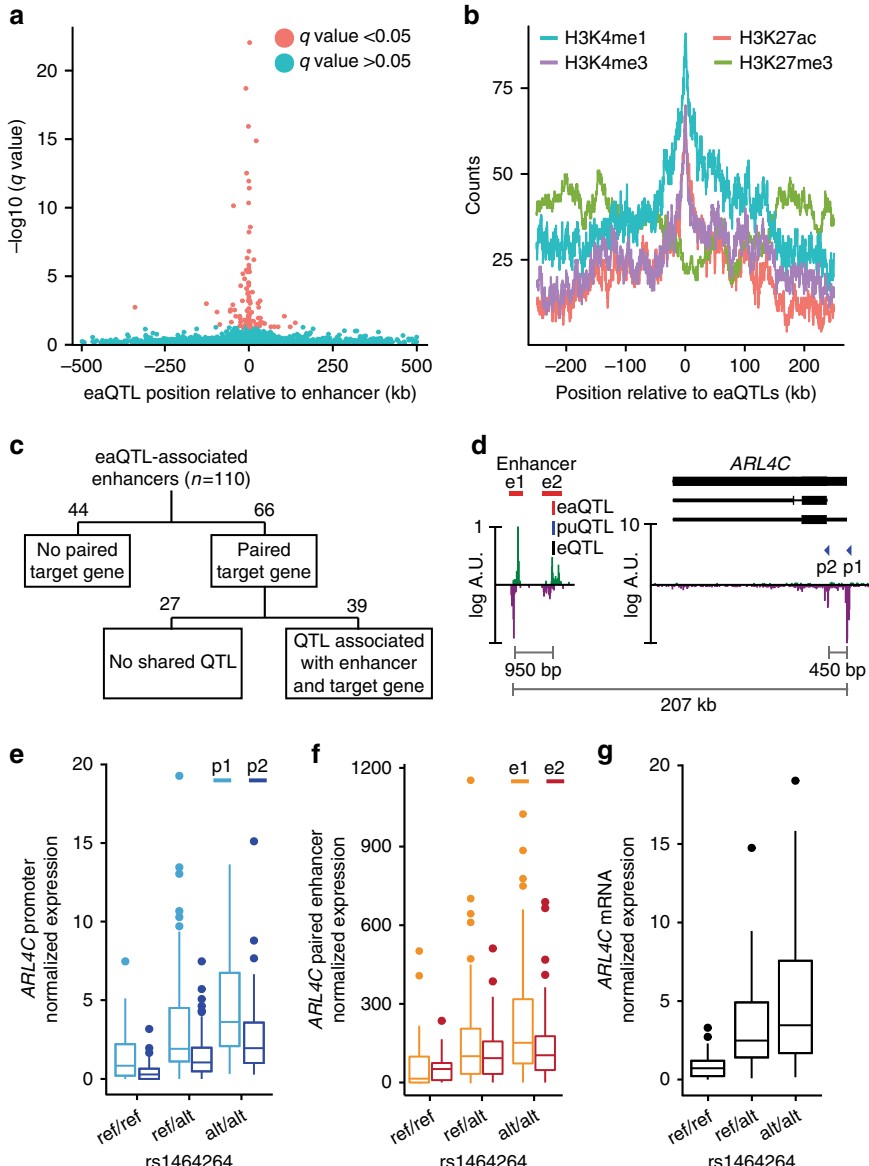

**Fig. 4** eaQTLs mapping and integration with puQTLs and eQTLs. **a** Significance relative to distance for each enhancer best-associated eaQTL plotted for a 1 Mb window. **b** Total counts of histone marks ChIP-seq signals (GM12878 cells, ENCODE data) for the 500 kb regions flanking eaQTLs. **c** Schematic representation of the procedure to identify triplets of regulatory variants, enhancers, and paired promoters. **d** CAGE signal at the *ARL4C* locus and associated enhancer region. The variant (rs1464264) mapped as a puQTL for *ARL4C*-associated CAGE peaks (p1, p2) and eaQTL for paired enhancers (e1, e2) is shown. Normalized promoter (**e**), enhancer (**f**), and mRNA (**g**) expression relative to each genotype group are plotted for the entire population (n = 154)

We hypothesized that not all puQTLs have an effect on the total mRNA production from a gene and that there exists either compensatory or antagonist mechanisms among different promoters. We addressed this question by estimating the fraction of puQTLs that are also associated with total mRNA levels, measured from RNA-seq for 154 individuals[4,5] (Supplementary Table 3). We measured the enrichment of low $p$ values according to the proposal of the Geuvadis consortium[5]. Briefly, we applied the $\pi_1$ statistics[27] to estimate the proportion of truly alternative features for the 2894 puQTLs associated with genes for which mRNA levels were quantified. We find 83% of the puQTLs being also significant eQTLs (Supplementary Fig. 5). This proportion decreases to 68% for the puQTLs that have opposite effects on multi-promoter genes (group-3), while it ranges between 81 and 88% for the other groups (Fig. 2g). We concluded that about a third of group-3 genes, whose total mRNA levels do not

significantly vary within the population, are associated with puQTLs triggering promoter shifts and therefore generating different isoform prevalence. An illustrative example is the *TTC23* gene (ENSG00000103852.8) from group-3. *TTC23* is represented in our data set with three CAGE peaks, localized in two promoter regions 1.6 kb apart (Fig. 3a). The SNV rs8028374 was mapped as a puQTL with significant opposite effects on the two promoter regions (ß = 0.56 for p1 and ß = −1.03 for p2) (Fig. 3b). Notably, we did not find significant eQTL for the *TTC23* mRNA level in data from Stranger and colleagues[4] (Fig. 3c) or the GTEx consortium[6]. Additional examples include the *CD97* (ENSG00000123146.15) and *FAM76B* (ENSG00000077458.8) genes (Supplementary Fig. 6a, b). The partial shifts observed between promoters reveal a plasticity in promoter usage that is potentially implicated in preserving suitable steady-state mRNA levels.

One example of a multi-promoter gene for which promoter usage analysis provides a hypothetical mechanism underlying the effect of an associated eQTL is *DENND2D* (ENSG00000162777.12). The seven *DENND2D* CAGE peaks are distributed in three distinct promoters within a 4.3 kb region (Fig. 3d). We detected only one promoter region, with two CAGE peaks, significantly associated with a puQTL (rs35430374, $p$ value $= 2.54 \times 10^{-11}$ for p4 and $p$ value $= 1.50 \times 10^{-12}$ for p7, Fig. 3e). This signal was replicated using the expression values for exons specific of transcript isoforms produced from either of the three promoter regions (Supplementary Fig. 6c). Remarkably, the same variant is detected as an eQTL ($p$ value $= 1.73 \times 10^{-6}$) for total mRNA levels (Fig. 3f), which suggests that activation of an alternative promoter (here p4) results in the observed eQTL effect. Indeed, the CAGE peak p4 appears to be driving the largest fraction of the observed variance for the mRNA level. Similar cases were observed among puQTL associated with genes of group-3, such as *MCM8* (ENSG00000125885.9) and *TLR1* (ENSG00000174125.3) (Supplementary Fig. 6d, e).

Taken together, the integration of puQTLs and eQTLs provides new insights into the mechanisms underlying eQTLs, explaining the prevalence of transcript isoforms and the relative participation of alternative promoters to the transcriptional output of genes.

**Regulatory variants associated with enhancer activity.** Among the previously non-annotated CAGE peaks associated with puQTLs, the 213 located in enhancer regions (Fig. 1b), as defined by characteristic chromatin modification patterns[21], are TSSs of eRNAs[22]. Bidirectional-capped RNA production has been described as a hallmark of active enhancers and used to detect and measure enhancer activity in numerous human and mouse cell types[28,29].

We sought to use the CAGE transcriptome profiling to gain further insights into the implication of regulatory variants on enhancer regions, mapping variants associated with eRNA levels, considered here as a proxy of enhancer activity.

Following a comparable approach to that used for promoter usage, we mapped the CAGE tags to the FANTOM enhancer atlas elements[28] and quantified the expression of 3558 transcriptionally active enhancers. We performed *cis*-QTL mapping for enhancer activity using TADs as *cis*-windows, mapping 110 enhancer activity-QTLs (eaQTLs) associated with enhancer transcriptional activity at the significance threshold of 5% FDR (Supplementary Fig. 7a). eaQTLs are enriched in the proximity of associated enhancers (Fig. 4a), within open chromatin regions (Supplementary Fig. 7b), and in loci carrying histone marks specific of enhancer elements and active transcription (Fig. 4b). In accordance with a previous report for enhancers and promoters[30], higher H3K4me1 signals than H3K4me3 signals are observed with ChIP-seq at eaQTLs sites, while the opposite trend characterized puQTLs despite a fraction of them overlapping enhancer elements (Supplementary Fig. 7c).

We hypothesized that a combination of puQTL and eaQTL analyses may contribute to the identification of regulatory variants associated with gene expression, which effects are essentially exerted on enhancer regions. To address this question, we used the Enhancer-FANTOM Robust Promoter associations[28] to link 66 enhancers with 293 CAGE peaks in 322 unique eaQTL-enhancer–promoter triplets. Among these triplets, 39 include eaQTLs which are also mapped as puQTLs or in LD with a puQTL (Spearman's $\rho > 0.8$) (Fig. 4c). An illustrative example is the *ARL4C* gene (ENSG00000188042.5) (Fig. 4d). The rs1464264 variant was mapped as a puQTL for the two *ARL4C* promoters ($p$ value $= 1.64 \times 10^{-8}$ for p1 and $p$ value $= 5.33 \times 10^{-10}$ for p2)

(Fig. 4e), as an eaQTL for two enhancers ($p$ value $= 2.62 \times 10^{-7}$ for e1 and $p$ value $= 4.26 \times 10^{-5}$ for e2) paired with *ARL4C* (Fig. 4f) and as an eQTL ($p$ value $= 7.74 \times 10^{-7}$) (Fig. 4g). These observations allow us to build a hypothetical mechanism for the *ARL4C*-associated eQTL, which includes increased activity of the enhancer located ~200 kb downstream of the *ARL4C* promoter. The higher level of *ARL4C* mRNA observed in the presence of the alternative allele state is, under this hypothetical model, driven by the increased activity of the distant enhancer region resulting from the genetic variation. A similar scenario was observed for the *SWAP70* gene (Switching B-cell complex subunit 70, ENSG00000133789.10) and an enhancer region localized 50 kb upstream of it (Supplementary Fig. 7d–g).

Finally, for the set of 39 regulatory variants identified as both eaQTLs and puQTLs, we assessed causal molecular relationships for model networks including (1) the eaQTL variants, (2) the enhancer transcriptional activity, and (3) the paired promoter expression values. Using causal inference testing (*cit* R package)[31] for 92 triplets (39 eaQTL-associated enhancers paired with 78 promoters), we tested independently two models that considers enhancer transcriptional activity as the molecular mediator of gene expression and vice versa (Supplementary Fig. 8a). The effect of regulatory variants, mapped as puQTLs and eaQTLs, on enhancer activity was found causal for the association observed with the target gene expression level for 17 triplets at the significance threshold of 5% FDR, involving 12 eaQTLs (Supplementary Fig. 8b).

Taken together, we provide here the first set of human regulatory variants associated with enhancer activity based on eRNA quantification and illustrate the potential of using complementary molecular phenotypes to dissect the mechanism(s) underlying enhancer related eQTLs.

## Discussion

We described here a collection of 5376 puQTLs and 110 eaQTLs; regulatory variants associated with promoter usage and enhancer activity, respectively. By levering the CAGE technology to quantify these molecular phenotypes, this study highlights how CAGE transcriptome profiling coupled with QTL mapping can help dissect the genetic mechanisms underlying eQTLs and potentially disease-associated variants.

We find extensive overlap between puQTLs mapped from CAGE data and eQTLs mapped from RNA-seq, as a result of the expected high correlation between the mRNA quantification provided by both technologies[32]. While the analysis of exon usage with RNA-seq requires increased sequencing depth[33], it enables an exon-wise quantification that complements the specific TSS information provided by CAGE. The combination of the two approaches (i.e., RNA-seq and CAGE *cis*-QTL mapping) therefore constitutes an effective strategy to give a broader view of the molecular mechanisms underlying regulatory variants. We leveraged these advantages to discover puQTLs involved in differential promoter usage and by extension to differential transcript isoform production. Our approach identified, among genes of groups-2, -3, and -5 (Fig. 2), regulatory processes for 5′-end transcript variations adding on the current knowledge of alternative transcript-splicing QTLs[5,7,8]. While we opted to analyze the effects of genetic variation on differential promoter usage, a recent study has mapped regulatory variants associated with the TSS usage (tssQTLs) by performing single nucleotide resolution TSS phenotyping in fruit-flies using CAGE[34]. They describe tssQTLs not affecting transcript levels, in line with our observations.

The low expression level and poor annotation of lncRNAs[35] limit the power to identify lncRNA-eQTLs[36]. Nevertheless, eQTL

studies on lncRNAs, even restricted to a few hundred non-coding genes, established a substantial contribution of lncRNA-associated regulatory variants to human phenotypes[5,36–39]. As anticipated by a recent report[40], and revealed in our study with the detection of puQTLs associated with 489 lncRNAs and 271 antisense transcripts, the precise genome-wide TSS mapping and the accurate quantification of low-expressed non-coding transcripts are complementary features that CAGE can provide for conducting QTL mapping, compared to RNA-seq. Moreover, our strategy could further contribute to the characterization of potential roles for lncRNAs in human traits, by combining it with the recently produced atlas of putative functional human lncRNA[41]. This atlas has over 9000 lncRNAs, including about 3000 enhancer-associated lncRNAs.

Evidence supporting the so called "multiple enhancer variant" hypothesis for GWAS traits has been reported for loci carrying multiple regulatory variants within enhancers and cooperatively altering the expression of target genes[42]. Although high-throughput reporter assays have been used to test regulatory consequences of non-coding variants on reporter genes[11,43,44], the lack of native chromatin context represents the main limitation of these methods that do not investigate epistatic interactions between multiple variants. The approach developed in our study to map eaQTLs constitutes a potential strategy to unravel regulatory mechanisms involving multiple variants within enhancer elements.

Overall, this study reveals that differential promoter usage is an important consequence of functional variation in the human genome. Our eaQTL mapping analysis provides the opportunity to dissect mechanisms underlying regulatory variants located within enhancer elements. Finally, our study highlights how CAGE transcriptome profiling coupled with QTL mapping furthers our understanding of eQTL effects and contributes to the effective interpretation of disease-associated variants.

## Methods

**Cell culture.** EBV-transformed LCLs (Supplementary Table 1) purchased from the Coriell Cell Repository (CEU, $n = 86$, with the authorization of the ethical committee of the University of Geneva Medical School) or from the GenCord collection ($n = 68$, informed consent was obtained from all human subjects and the project approved by the local ethics committee at the University Hospital of Geneva CER 10-046)[45] were cultured in conventional medium for LCLs (RPMI 1640, GlutaMAX; Gibco) with 15% fetal bovine serum (Gibco), 50 U ml$^{-1}$ penicillin and 50 μg ml$^{-1}$ streptomycin (Gibco). While harvesting cells in growing phase ($<10^6$ cells ml$^{-1}$), culture media were systematically tested for mycoplasma infection (Venor GeM Mycoplasma detection kit, Sigma-Aldrich) prior to proceeding with RNA extraction.

**RNA preparation.** For each cell line, a nucleus-enriched RNA fraction was isolated from 20 million cells, as detailed in Fort and colleagues[46]. Briefly, cells were first lysed in chilled lysis buffer (0.8 M sucrose, 150 mM KCl, 5 mM MgCl$_2$, 6 mM 2-mercaptoethanol, and 0.5% NP-40) and centrifuged for 5 min at 10,000×g (4 °C). Nuclei pellets were washed twice with lysis buffer before resuspension in TRIzol Reagent (Life Technologies). The RNeasy kit (Qiagen) was used according to the manufacturer's protocol to extract nucleus-enriched RNA fractions. During the RNA purification process, samples were treated with DNase I (TURBO DNA-free kit, Ambion) following the manufacturer's recommendations.

**CAGE library preparation and data processing.** CAGE libraries were prepared from 3 μg of RNA, using the reagents and following the protocols published by Takahashi and colleagues[47]. Briefly, the initial reverse transcription was performed using random primers and in the presence of sorbitol and trehalose. Then, the enrichment of capped RNAs was obtained with initial oxidation of the 5′-cap RNA diol group, resulting in a dialdehyde that was then coupled with long-arm biotin hydrazide before capture of biotinylated RNA/cDNA hybrids on streptavidin-coated magnetic beads. Samples were treated with RNase-I, cleaving single-stranded RNA and discarding cDNA that did not reach the 5′-cap. Finally, RNA/cDNA hybrids were denatured with alkali to recover cap-selected single-stranded cDNAs. Sample multiplexing was achieved by introducing barcode sequences within 5′-linkers that were ligated to the 3′ extremity of first-strand cDNA. The 5′-linkers were also used for priming the second-strand cDNA synthesis and for

CAGE tag generation using the EcoP15I restriction enzyme. Following 3′-linker ligation and prior to loading on the sequencing platform, a final CAGE library amplification using nine PCR cycles was performed.

CAGE libraries were sequenced on the Illumina HiSeq 2500 platform with a read length of 50 bases. Sequences with ambiguous base calling were discarded, samples reads split by barcodes and artefactual linker/adapter sequences removed using TagDust (v2.2)[48]. Reads were of 26–42 bases in length. CAGE tags were mapped to the reference genome hg19/GRCh37 using Delve (V1.0)[49] and Burrows-Wheeler Aligner (BWA V0.5.6)[50]. Two mismatches were allowed for the mapping procedure and only reads with MapQ values over 20, and therefore mapping to single loci of the reference genome, were used in our analyses. Finally, reads mapping to ribosomal RNA were eliminated.

Of the 164 samples originally sequenced, 8 with less than $5 \times 10^6$ mapped CAGE tags were discarded, in line with sequencing depth recommendations for the CAGE technology[47].

**Promoter expression.** Genomic coordinates of 195,296 robust autosomal human FANTOM CAGE peaks and their gene assignment annotations were retrieved in May 2016 from http://fantom.gsc.riken.jp/5/data/. CAGE tag counts per CAGE peak were normalized for sequencing depth, converting tag counts to tags per million mapped reads (TPM) and, similarly to the FANTOM promoter atlas[9], TPM values were further normalized between samples using the relative logarithmic expression (RLE) normalization procedure from edgeR[51]. We applied a minimum expression threshold on the mean expression over all individuals included in the study of 0.5 RLE-TPM (Supplementary Fig. 1b), and constructed an expression matrix including 38,759 autosomal CAGE peaks for the 154 individuals.

**Genotype data.** GenCord individuals were genotyped with the Illumina 2.5M Omni chip in a previous study[15]. Variants were imputed into 1000 Genomes phase-3[14] using SHAPEIT2 (V2.20)[52] and IMPUTE2 (V2.3.2)[53], yielding $9.1 \times 10^6$ SNVs. The genotyping data for the CEU individuals included in our study were retrieved from the whole-genome sequencing analyses performed by the 1000 Genomes Project Consortium[14]. We combined these two data sets and filtered for an information score above 0.5 and a minimum alternative allele counts of 10. We were left with genotype data at 7,508,202 autosomal variants for the 154 individuals.

**Genotype sequencing data consistency control.** Allelic consistency between genotype and CAGE tag sequences was assessed with the match BAM to VCF (MBV) methods (QTLtools package V1.1)[16]. Of the 156 samples passing the sequencing depth threshold, no amplification bias was detected and samples from two individuals, with suspicion of cross-sample contamination, were removed from the study (Supplementary Fig. 1c).

**Mapping QTLs.** We mapped puQTLs and eaQTLs using QTLtools (V1.1)[17], with the following sets of covariates: for the puQTLs, the first 3 PCs were derived from genotype data and the first 20 PCs were derived from promoter-normalized expression values. We controlled for stratification due to sample collections (Supplementary Fig. 2a) and library preparation batches (Supplementary Fig. 2b) in the normalized promoter usage data. For the eaQTL mapping, we used the first 3 PCs derived from genotype data and the first 12 PCs derived from enhancer-normalized activity values.

We delimited the set of variants to be tested per molecular phenotype by using TADs, as defined by Hi-C data on LCLs[19]. To determine whole-genome significance, first 1000 permutations were performed to adjust nominal $p$ values for the number of independent tests performed for each promoter or enhancer per cis-window. Second, adjusted $p$ values were corrected for the total number of promoters or enhancers tested genome-wide using the $q$ value R package (V2.2.2)[54]. We finally extracted puQTL or eaQTL with $q$ value <0.05, which corresponds to a 5% FDR.

**Enrichment analysis.** The QTLtools fenrich module (V1.1)[17] tests if a set of QTLs fall within functional annotations more often than is expected by chance. We used this module with annotations retrieved from the UCSC Table Browser on September 2016. These data are available for the GM12878 LCL, including 75 ChIP-seq experiments for four histone marks (H3K4me1, H3K4me3, H3K27ac, and H3K27me3) and 71 transcription factors (ENCODE uniform peaks)[23] as well as DNase I hypersensitivity (ENCODE/University of Washington). 1000 permutations were performed on the functional landscape, here all promoters detected ($n = 38,759$), to obtain expected overlap values.

**GWAS hits enrichment.** To assess how many puQTLs are overlapping GWAS variants, we selected puQTLs either matching the variants reported in the GWAS catalog[24] or near to a GWAS variant (±500 kb) with a R-squared greater than 0.5. To estimate the expected overlap by chance of puQTLs and GWAS hits, we performed 1000 permutations using random variants with allelic frequencies and distances to TSSs that are similar to those of puQTLs.

The RTC analysis was performed using the *QTLtools rtc* module (V1.1)[17] using the GWAS catalog[24] (accessed in May 2017).

**Classification of puQTL-associated genes**. We initially grouped puQTL-associated genes with either a single CAGE peak or with several CAGE peaks when less than 200 nt apart. The other multi-promoter genes were categorized based on the effect of puQTLs on their different CAGE peaks. To this end, the effect size and associated *p* value for each CAGE peak of corresponding puQTL-associated genes were calculated, and puQTL-promoter associations with a *p* value <0.05 were considered. We grouped genes associated with puQTLs having significant concordant regulatory effects (ß, regression slope) into group-4 and group-5 based on the effect size ratio (ER = |max(ß)|/|min(ß)|), grouping genes with different effects on CAGE peaks when ER > 2.

**RNA-seq mRNA quantifications and eQTL mapping**. We retrieved mapped RNA-seq data (BAM files) for 154 Central European individuals[4,5] (Supplementary Table 2). We performed gene and exon quantification using *QTLtools quant*[17] and GENCODE-V19 as the reference for transcripts. Genotype data for these samples were retrieved from whole-genome sequencing analyses performed by the 1000 Genomes Project Consortium[14]. We used *QTLtools cis* (V1.1)[17] module with nominal pass, gene expression matrixes and, as set of covariates, we used the first 3 PCs derived from genotype data and the first 20 PCs derived from gene-normalized expression values.

**TSS annotation**. The CAGE peaks retrieved from the FANTOM promoter atlas were annotated using the GENCODE-V19[20] transcripts reference set and the ChromHMM[21] segmentation based on ENCODE ChIP-seq histone marks from the LCL GM12878. Our hierarchical annotation procedure has four steps. First, FANTOM CAGE peaks within 500 nt upstream of annotated TSSs or residing within a 5′-UTR first exon or a 5′-UTR first intron were annotated as "Annotated genes". The other CAGE peaks were annotated as "Not annotated transcript" and further categorized in "enhancer," "promoter," or "other" based on epigenetic features. Finally, the "promoter" was subdivided into "Anti-sense promoter" and "Putative promoter" based on genomic localization.

**Enhancer activity quantification**. Enhancer regions transcriptionally active in our cohort of LCLs were selected with a procedure similar to the approach detailed in the FANTOM enhancer atlas[28,] where they detected enhancers based on balanced bidirectional transcriptional hallmarks[22]. First, 63,991 autosomal enhancer regions were retrieved in May 2016 from the FANTOM atlas (http://fantom.gsc.riken.jp/5/data/). Enhancer elements characterized with bidirectional transcription patterns in our samples were selected. To this end, we first produced CAGE tag clusters using the *Paraclu* method (V9)[55], with CAGE tag 5′ genomic coordinates as input and (i) a minimum of five tags per cluster, (ii) a (maximum density)/(baseline density) ≥ 2 and (iii) a maximal cluster length of 200 nt. To select enhancer regions with a bidirectional transcriptional pattern, we required the overlap of two CAGE tag clusters on opposite DNA strands within a 400 nt window from the enhancer region center.

We then calculated normalized expression for both flanking 200 nt windows (F and R) to determine, for each enhancer region, a directionality score, D = (F−R)/(F + R). Counts of CAGE tags per F and R windows were normalized for sequencing depth, converting tag counts to tags per million mapped reads (TPM) and, similarly to FANTOM[9], TPM values were further normalized between samples using the RLE normalization procedure from *edgeR*[51]. We then filtered enhancer regions to have non-promoter-like directionality pattern, requiring |D| < 0.8. Finally, we summed the twice normalized expression of the 200 nt flanking windows to assign a single expression value to each enhancer, and discarded enhancers with null expression in more than 50 individuals. We built an expression matrix for 3558 enhancers for 154 individuals.

**Data availability**. The sequencing CAGE data generated in this study are available in the ArrayExpress database at EMBL-EBI (www.ebi.ac.uk/arrayexpress) under accession number E-MTAB-5835. Derived data supporting the findings of this study are available from the corresponding author on request.

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

## Acknowledgements

This work was supported by a Swiss National Science Foundation Ambizione Grant (PZ00P3-154728 to A.F.). The computations were performed at the Vital-IT (http://www.vital-it.ch) Centre for High-Performance Computing of the Swiss Institute of Bioinformatics. The authors thank Cedric Howald, Corinne Gherig, and Michel Guipponi for computational and laboratory assistance.

## Author contributions

A.F. coordinated the project, laboratory data production, data analyses, and writing. A.F. and D.M. produced the CAGE libraries. P.C. provided reagents and assistance for technology transfer. A.F., M.G. and O.D. performed the analyses. O.D., F.S., E.T.D. and S.E.A. contributed to the design and interpretation of the analyses. R.J.F. contributed to the writing of the manuscript. All authors commented on the interpretation of results, reviewed, and approved the final manuscript.

## Additional information

**Competing interests:** The authors declare no competing financial interests.

