## [Peer Review File · Nature Communications]

Reviewer #1 (Remarks to the Author):

In this study the authors generate CAGE data from a panel of 154 lymphoblastoid cell lines. They use this data to identify genetic variants associated with promoter usage (puQTLs) and enhancer usage (eaQTLs). The dataset will be a nice resource that complements eQTLs and expands the set of molecular QTLs that are available for study. The methods that the authors use are fairly standard and utilize tools that have recently been developed for the study of eQTLs. Most of the analyses are solid and figures provide a nice presentation of the results.

There are a few issues and weaknesses with the paper:

1)

L104-111. The authors look at the concordance between puQTLs and GWAS hits from the NHGRI GWAS catalog and identify a total of 35 overlaps. The analysis is somewhat superficial, however, and it's not clear how to interpret the result. For example, it is not clear if 35 is a significant enrichment over what would be expected by chance. Furthermore, for many traits the GWAS catalog only contains lead SNPs or the subset of SNPs that are present on the SNP chip used in the study. It doesn't seem like the authors fill in non-genotyped sites (e.g. using imputation). For these reasons this analysis does not seem very strong, and I suggest that they either leave it out or conduct a much more thorough analysis.

2)

L148: The authors use the π_1 statistic which gives the estimated fraction of tests for which the data are from the alternative hypothesis. I think this is a reasonable use of the statistic, but the authors need to explain it better and provide a citation for how it is determined, because I do not believe that most readers will know this.

3)

Figure 1d: Panel D of Figure 1 is very confusing, and I am not sure what interpretation(s) should be drawn from it. While it is nice to have a summary of the characteristics of puQTLs, it seems strange to present classification/annotation of just the subset of puQTLs that are NOT linked not with promoters in FANTOM. Wouldn't it make more sense to present the classifications of all of the puQTLs, not just the subset that are not linked with promoters in FANTOM? In addition the classifications are counterintuitive and some of them feel like they don't belong together in the same plot. E.g. a puQTL in a putative promoter doesn't seem like it should also be intergenic or intronic. Also, it might be better to show the fraction of antisense/sense puQTLs in a separate plot.

4)

Figure 4b: The eaQTLs appear to have high H3k4me3 on average, which is more consistent with enhancers than promoters. If CAGE peaks overlapping H3K24me3 peaks are excluded as putative enhancers, how many eaQTLs are left?

5)

The discussion could be improved substantially. There are numerous confusing sentences such as those at lines L251-254, L258-260. I do not understand the substantial focus on lncRNAs in the discussion since lncRNAs are not really mentioned elsewhere in the paper. The discussion mentions epistatic interactions (L280) and says that this study is "a proof-of-principle for a strategy with the potential to unravel regulatory mechanisms involving multiple variants within enhancer elements" but I don't feel that this is something that is provided by this paper and don't agree with this statement.

Reviewer #2 (Remarks to the Author):

The authors present a study of eQTLs at the promoter and enhancer level, using CAGE profiling of lymphoblastoid cell lines in a large cohort of subjects in order to associate promoter/enhancer expression to genetic variation. They identify 5376 eQTLs at the promoter level, some of which have opposite effects on different promoters of the same gene. In addition, they find 110 eQTLs that appear to modulate enhancer activity as measured by bidirectional expression at putative enhancer loci. Finally, they attempt to link enhancers and promoters together and identify casual links between genetic variants, enhancer expression and promoter expression. The paper is of great interest for the study of the influence of genetic variation on gene regulation through promoter and enhancer interplay.

The manuscript is well written and easy to follow, with conclusions that are mostly well founded. However, in some cases it's not clear that the conclusions are fully supported by the data. Specifically, the last part of the analysis (linking enhancers to promoters, and establishing causal links between eQTLs and enhancer/promoter expression) is weak and either needs a more rigorous analysis, or the conclusions need to be downplayed.

Major points:

1. The authors use linear regression to couple enhancers with promoters within the same TAD based on their expression, and a p-value cutoff of 0.01. However, this p-value should be corrected for multiple testing, it is not clear whether it has. Additionally, even though the number of samples are fairly large (~150), expression correlation is a poor way of identifying true interactions between enhancers and promoters. Using physical interaction data (e.g. Hi-C or Pol2 mediated ChIA-pet data available from ENCODE for the same and similar cell lines, https://www.encodeproject.org/search/?type=Experiment&assay_title=ChIA-PET) would be much stronger. Also, another (albeit also expression based) way of finding putative enhancer-promoter pairs is to look at correlation across the much larger sample collection in FANTOM5 (<http://enhancer.binf.ku.dk/presets/> under heading "5. Enhancer - FANTOM Robust Promoter associations").
2. "Casual inference testing" is used to assign casual relationships between genetic variant, enhancer expression and promoter expression. Again it is not clear whether the p-value cutoff has been properly adjusted for multiple testing correction.

Minor points:

3. Comparing Figure 1C and 4B, it looks like the enrichment of histone modifications around enhancer and promoter eQTLs are quite similar - it's a bit difficult to compare since the data for enhancers is more noisy. In the figure legends for these figures it says that "Density of histone marks" is being displayed, however the y-axis says "counts" - what is meant by this? Has the data been normalized? Also, what cell type is this histone data from? I assume the ENCODE GM12878 data but it should be explicitly stated in the figure legend.
4. Although the language in the article is generally acceptable, there are numerous minor grammatical errors that would easily be caught by a check from a native English speaker. One example:
P7 L194: Following comparable approach than for promoter usage ->
Following a comparable approach as for promoter usage

Reviewer #3 (Remarks to the Author):

The authors have done a nice job with this analysis to identify regulatory variants associated with promoter usage and enhancer activity in EBV-transformed lymphoblastoid cells. The statistical analyses could be described better in some places, but overall appear to have been performed with high quality and the findings have been interpreted appropriately. One limitation is that conclusions are based predominantly on statistical analysis and associations identified from large-scale datasets, without the addition of definitive wet lab experiments to conclusively demonstrate regulatory activity at some of the identified loci. A possible concern is that the 154 samples do not appear to have been filtered for quality control which may raise questions concerning the influence of an outlying sample that is apparent in Supplementary Figure 2. Finally, some of the language was awkward and the manuscript could probably be improved by further editing. In my review I have provided several suggestions that the authors may consider in a potential revision of their manuscript in order to clarify certain aspects of the statistical and bioinformatic methods.

(1) The study has investigated EBV-transformed lymphoblastoid cell lines. However, this cell type is not indicated in the title or the current Abstract. Since the effects of gene regulatory variants seem likely to be at least partially (if not strongly) cell type-specific, this would be important information to include in the Abstract and potentially the title as well.

(2) The authors do not appear to have applied quality filtering among the 154 samples and have included all 154 available samples in the analysis. This may be appropriate, although for example in Supplemental Figure 1 it is shown that sequencing depth varies 6-fold among samples. Does removal of samples with lower sequencing depth improve identification of puQTLs and eaQTLs associated with weakly expressed transcripts?

(3) Likewise, in supplemental Figure 2, there is a sample shown in (A) and (B) that appears to be an outlier. Was the possibility of removing this sample considered? How sensitive are results to these potentially outlying values?

(4) "We applied a minimum expression threshold on the mean expression over all individual included in the study of 0.5 RLE-TPM." This procedure is reasonable, but it would be helpful to describe the justification of the 0.50 threshold. Could values above this threshold still be attributed to background noise? How does the value of 38,759 autosomal CAGE-peaks vary with changes in the RPE-TPM threshold applied?

(5) The study combines data from two different sets of samples, including the Coriell Cell Repository (CEU, n=86) and the GenCord collection (n=68). The samples do not appear to show systemic differences with regard to gene expression based upon principal component analyses, but was it possible to identify individual genes with significantly different expression between the two groups of samples?

(6) Genotypes were imputed for data generated using the Illumina 2.5M Omni chip. Was there a particular r^2 value applied to ensure the imputed genotypes were of good quality?

(7) Lines 358 and 359. The authors state "No amplification bias was detected and individuals with suspicion of cross-sample contamination were removed from the study (Supplementary Fig. 1b)." It would be helpful to indicate here how many individuals were removed due to possible cross-sample contamination.

(8) Lines 372 – 374. The authors state "To determine whole genome significance, corrected p -

values for multiple variants within cis-windows were corrected for multiple promoters or enhancers being tested, using the qvalue R package (V2.2.2)". It would be helpful for the authors to expand on this. Does this approach require that the p-values are independent?

(9) It would be appropriate to include additional information regarding the 154 samples used in the analyses, in particular the sex of the samples. Was the number of males and females approximately balanced?

(10) Supplemental Figure 3, part D. Are confidence intervals or standard errors shown for each transcription factor? It would be helpful to clarify this in the figure legend and possibly also indicate using a different color which odds ratios are significant.

Reply to Reviewers' comments for the manuscript NCOMMS-17-15061

We thank the Reviewers for their careful work on our manuscript and for highlighting various points that needed to be corrected or modified. Based on their remarks, we have reworked the appropriate paragraphs (highlighted in red in the revised manuscript) and figures and performed additional analyses. We appreciate their constructive comments, which have helped us to improve substantially the manuscript.

The main modifications concern additional enrichment analyses for overlap between puQTLs and GWAS hits as well as in depth revision of the Discussion section.

Reviewer #1 (Remarks to the Author):

In this study the authors generate CAGE data from a panel of 154 lymphoblastoid cell lines. They use this data to identify genetic variants associated with promoter usage (puQTLs) and enhancer usage (eaQTLs). The dataset will be a nice resource that complements eQTLs and expands the set of molecular QTLs that are available for study. The methods that the authors use are fairly standard and utilize tools that have recently been developed for the study of eQTLs. Most of the analyses are solid and figures provide a nice presentation of the results.

We thank the Reviewer for her/his positive comments on the manuscript and for recognizing the originality of this study as well as the value of the resource for the eQTLs field.

There are a few issues and weaknesses with the paper:

1)

L104-111. The authors look at the concordance between puQTLs and GWAS hits from the NHGRI GWAS catalog and identify a total of 35 overlaps. The analysis is somewhat superficial, however, and it's not clear how to interpret the result. For example, it is not clear if 35 is a significant enrichment over what would be expected by chance. Furthermore, for many traits the GWAS catalog only contains lead SNPs or the subset of SNPs that are present on the SNP chip used in the study. It doesn't seem like the authors fill in non-genotyped sites (e.g. using imputation). For these reasons this analysis does not seem very strong, and I suggest that they either leave it out or conduct a much more thorough analysis.

We thank the Reviewer for pointing out this important point that we addressed by performing an additional analysis and providing further details regarding our methodology.

To reinforce the point that puQTLs are enriched within GWAS regions, we calculated the enrichment of puQTLs and GWAS region overlap compared to the overlap expected by chance. The results of this analysis are detailed in the Results section:

“We then estimated the significance of the overlap between puQTLs and disease-associated variants reported in the GWAS catalog²⁴. We found 1,024 puQTLs (out of 4,876 unique variants) overlapping linkage disequilibrium (LD) intervals containing GWAS hits (OR: 1.25, CI: 1.16-1.35, p-value=0.0009). Such significant enrichment has been previously reported for QTLs associated with other molecular phenotypes (eQTLs^{5,6}, splicing QTLs⁷, methylation QTLs²⁵).” (L.116-121).

Details about the approach we used are mentioned in a new paragraph of the Methods section:

“GWAS hits enrichment

To assess how many puQTLs are overlapping GWAS variants, we selected puQTLs either matching the variants reported in the GWAS catalog²⁴ or near to a GWAS variant (+/- 500 kb) with a R-squared greater than 0.5. To estimate the expected overlap by chance between puQTLs and GWAS, and thereby measure the significance of the observed overlap, the same number of random variants as tested puQTLs, with similar allelic frequencies and distribution of distances to TSSs, have been tested over 1,000 permutations.” (L.406-413)

We agree with the Reviewer that non-genotyped sites cannot be tested, but from the 40,530 GWAS catalog entries, 35,776 (88%) are present in our reference.

To improve clarity, we now provide further details regarding the RTC analysis, which we feel better explains the extent and deepness of the approach that allows assessment of whether effects of the overlapping puQTLs and GWAS variants are tagging the same functional variants.

Current version: *“To refine the analysis and decipher whether the effect of the puQTLs and GWAS variants are concordant (i.e. tagging the same functional variant), we applied the Regulatory Trait Concordance (RTC)²⁶ method. Briefly, RTC accounts for local LD structure and regresses out the genetic effect of the GWAS variant from the CAGE peak expression data, to measure if the puQTLs association is still significant. The RTC scores range between 0 and 1, reflecting not concordant or concordant effects of the pair of puQTL and the GWAS hit, respectively. We found 51 puQTLs passing the high-confidence concordance threshold (RTC>0.9, Supplementary Table 2) for 35*

GWAS hits associated with a variety of phenotypic traits including systemic lupus erythematosus and inflammatory bowel disease. This finding proves the potency of our approach to detect functional variants implicated in disease etiology.” (L.121-132)

2)

L148: The authors use the π_1 statistic which gives the estimated fraction of tests for which the data are from the alternative hypothesis. I think this is a reasonable use of the statistic, but the authors need to explain it better and provide a citation for how it is determined, because I do not believe that most readers will know this.

We agree with the Reviewer that further explanations and references are needed here. We now mention the reference for the R package *qvalue* (Storey J.D. & Tibshirani R., *PNAS*, 2003) that described in detail the use of π_1 statistics, and we also clarify the approach in the main text.

Current version: *“We addressed this question by estimating the fraction of puQTLs that are also associated with total mRNA levels, measured from RNA-seq for 154 individuals^{4,5} (Supplementary Table 3). We measured the enrichment of low p-values according to the proposal of the Geuvadis consortium⁵. Briefly, we applied the π_1 statistics²⁷ to estimate the proportion of truly alternative features for the 2,894 puQTLs associated with genes for which mRNA levels were quantified.” (L.156-161).*

3)

Figure 1d: Panel D of Figure 1 is very confusing, and I am not sure what interpretation(s) should be drawn from it. While it is nice to have a summary of the characteristics of puQTLs, it seems strange to present classification/annotation of just the subset of puQTLs that are NOT linked not with promoters in FANTOM. Wouldn't it make more sense to present the classifications of all of the puQTLs, not just the subset that are not linked with promoters in FANTOM? In addition and the classifications are counterintuitive and some of them feel like they don't belong together in the same plot. E.g. a puQTL in a putative promoter doesn't seem like it should also be intergenic or intronic. Also, it might be better to show the fraction of antisense/sense puQTLs in a separate plot.

We thank the Reviewer for raising this point that we agree could be confusing to readers.

We first want to clarify that the former Fig. 1d (now Fig. 1b) describes the annotation of the CAGE peaks associated with puQTLs and not the annotation of the puQTLs themselves.

To gain in clarity, we now report only the CAGE peaks annotation based on GENCODE-

V19 and no longer the FANTOM original annotation based on GENCODE-V12. We also modified the paragraph and Fig. 1b describing the chromatin features and genomic position of the puQTL-associated CAGE peaks.

Current version: *“These puQTLs consist of 4,876 unique regulatory variants associated with 5,376 CAGE peaks (read promoters) assigned to 2,697 protein-coding and 489 non-coding genes, as well as 849 putative novel transcripts. We then combined the human transcript catalog (GENCODE-V19)²⁰ and histone marks profiling²¹ to annotate the puQTL CAGE peaks not associated with genes. The majority of these carry histone marks characteristic of promoter regions (n=515) and can be classified as either antisense promoters (n=271) or putative promoters localized within a gene or in an intergenic region (n=244) (Fig. 1b). Interestingly, 227 CAGE peaks carry histone marks characteristic of enhancer regions and thus can be considered as enhancer-RNAs (eRNAs)²².”* (L.93-102).

The “TSS annotation” method section was modified accordingly.

“TSS annotation

The CAGE peaks retrieved from the FANTOM promoter atlas were annotated using the GENCODE-V19²⁰ transcripts reference set and the ChromHMM²¹ segmentation based on ENCODE ChIP-seq histone marks from the LCL GM12878. Our hierarchical annotation procedure has four steps. First, FANTOM CAGE peaks within 500 nt upstream of annotated TSSs or residing within a 5'-UTR first exon or a 5'-UTR first intron were annotated as “Annotated genes”. The other CAGE peaks were annotated as “Not annotated transcript” and further categorized in “enhancer”, “promoter” or “other” based on epigenetic features. Finally, the “promoter” was subdivided into “Anti-sense promoter” and “Putative promoter” based on genomic localization.” (L.438-448).

4)

Figure 4b: The eaQTLs appear to have high H3k4me3 on average, which is more consistent with enhancers than promoters. If CAGE peaks overlapping H3K24me3 peaks are excluded as putative enhancers, how many eaQTLs are left?

We agree that further explanation regarding our interpretation of this observation are needed. We have now highlighted the differences in H3K4 methylation status between puQTLs and eaQTLs as well as at the associated CAGE peaks in a new Supp. Fig. 7c, and modified the Results section adding the following sentence:

“In accordance with a previous report for enhancers and promoters³⁰, higher H3K4me1 signals than H3K4me3 signals are observed with ChIP-seq at eaQTLs sites, while the opposite trend characterized puQTLs despite a fraction of them overlapping enhancer elements (Supplementary Fig. 7c).” (L.215-218).

To answer the second point of the Reviewer, we analysed in detail the ChIP-seq signal of each of four histone marks at CAGE peaks associated with eaQTLs. 22 out of the 110 of these appear to have non-null H3K27me3 signal, and 21 of these 22 putative enhancers present high signals for two or three active histone marks (data not shown). We thus conclude that a depletion of the repressive marks is characterising the associated putative enhancers.

5)

The discussion could be improved substantially. There are numerous confusing sentences such as those at lines L251-254, L258-260. I do not understand the substantial focus on lncRNAs in the discussion since lncRNAs are not really mentioned elsewhere in the paper. The discussion mentions epistatic interactions (L280) and says that this study is "a proof-of-principle for a strategy with the potential to unravel regulatory mechanisms involving multiple variants within enhancer elements" but I don't feel that this is something that is provided by this paper and don't agree with this statement.

To address the Reviewer's first comment, we further emphasize, in the revised manuscript, the advantage of combining RNA-seq with the CAGE technology for the detection of regulatory variants associated with the expression of non-coding transcripts. The actual number of non-coding genes associated with puQTLs as well as with CAGE peaks not linked with annotated genes are now detailed in the results section (see answer to point 3) and in Fig. 1b.

The discussion paragraph was modified as follows: "*The low expression level and poor annotation of lncRNAs³⁵ limit the power to identify lncRNA-eQTLs³⁶. Nevertheless, eQTL studies on lncRNAs, even restricted to a few hundred non-coding genes, established a substantial contribution of lncRNA-associated regulatory variants to human phenotypes^{5,36-39}. As anticipated by a recent report⁴⁰, and revealed in our study with the detection of puQTLs associated with 489 lncRNAs and 271 antisense transcripts, the precise genome-wide TSS mapping and the accurate quantification of low-expressed non-coding transcripts are complementary features that CAGE can provide for conducting QTL mapping, compared to RNA-seq. Moreover, our strategy could further contribute to the characterization of potential roles for lncRNAs in human traits, by combining it with the recently produced atlas of putative functional human lncRNA⁴¹. This atlas has over 9,000 lncRNAs, including about 3,000 enhancer-associated lncRNAs.*" (L.279-290).

We agree with the Reviewer's second comment and reworded our statement.

Current version: *"The approach developed in our study to map eaQTLs constitutes a potential strategy to unravel regulatory mechanisms involving multiple variants within enhancer elements."* (L.297-299).

Reviewer #2 (Remarks to the Author):

The authors present a study of eQTLs at the promoter and enhancer level, using CAGE profiling of lymphoblastoid cell lines in a large cohort of subjects in order to associate promoter/enhancer expression to genetic variation. They identify 5376 eQTLs at the promoter level, some of which have opposite effects on different promoters of the same gene. In addition, they find 110 eQTLs that appear to modulate enhancer activity as measured by bidirectional expression at putative enhancer loci. Finally, they attempt to link enhancers and promoters together and identify casual links between genetic variants, enhancer expression and promoter expression. The paper is of great interest for the study of the influence of genetic variation on gene regulation through promoter and enhancer interplay.

The manuscript is well written and easy to follow, with conclusions that are mostly well founded. However, in some cases it's not clear that the conclusions are fully supported by the data. Specifically, the last part of the analysis (linking enhancers to promoters, and establishing causal links between eQTLs and enhancer/promoter expression) is weak and either needs a more rigorous analysis, or the conclusions need to be downplayed.

We thank the Reviewer for recognizing the interest of this study and for emphasizing its relevance to the current knowledge on the influence of genetic variation on gene regulation through promoter-enhancer interplay.

Major points:

1. The authors use linear regression to couple enhancers with promoters within the same TAD based on their expression, and a p-value cutoff of 0.01. However, this p-value should be corrected for multiple testing, it is not clear whether it has. Additionally, even though the number of samples are fairly large (~150), expression correlation is a poor way of identifying true interactions between enhancers and promoters. Using physical interaction data (e.g. Hi-C or Pol2 mediated ChIA-pet data available from

ENCODE for the same and similar cell lines, https://www.encodeproject.org/search/?type=Experiment&assay_title=ChIA-PET) would be much stronger. Also, another (albeit also expression based) way of finding putative enhancer-promoter pairs is to look at correlation across the much larger sample collection in FANTOM5 (<http://enhancer.binf.ku.dk/presets/> under heading "5. Enhancer - FANTOM Robust Promoter associations").

We agree with the Reviewer that the use of the Enhancer - FANTOM Robust Promoter association is certainly more accurate than the expression correlation we performed on our dataset. Following her/his suggestion, we are reporting in the revised version of our manuscript enhancer-promoter pairs defined by the FANTOM consortium on its cell line samples. This modification does not change dramatically (39 instead of 38) the actual number of triplets (eaQTLs, enhancer, promoter) which include eaQTLs also mapped as puQTLs. However, as anticipated by the Reviewer, it reduces (92 instead of 158) the number of triplets that we test with causal inference testing.

We have modified the result section and panel Fig. 4c accordingly.

Current version: *"To address this question, we used the FANTOM robust enhancer-promoter association²⁸ to link 66 enhancers with 293 CAGE peaks in 322 unique eaQTL-enhancer-promoter triplets. Among these triplets, 39 include eaQTLs which are also mapped as puQTLs or in LD with a puQTL (Spearman's $\rho > 0.8$) (Fig. 4c)."* (L.221-225).

2. "Casual inference testing" is used to assign casual relationships between genetic variant, enhancer expression and promoter expression. Again it is not clear whether the p-value cutoff has been properly adjusted for multiple testing correction.

Following the modifications detailed in the preceding point, we now report 12 eaQTLs passing the statistical threshold of 5% FDR. The main text and Supplementary Fig. 8b have been modified accordingly and are now showing *q-values*.

Current version: *"The effect of regulatory variants, mapped as puQTLs and eaQTLs, on enhancer activity was found causal for the association observed with the target gene expression level for 17 triplets at the significance threshold of 5% FDR, involving 12 eaQTLs (Supplementary Fig. 8b)."* (L.244-248).

Minor points:

3. Comparing Figure 1C and 4B, it looks like the enrichment of histone modifications around enhancer and promoter eQTLs are quite similar - it's a bit difficult to compare since the data for enhancers is more noisy. In the figure legends for these figures it says

that "Density of histone marks" is being displayed, however the y-axis says "counts" - what is meant by this? Has the data been normalized? Also, what cell type is this histone data from? I assume the ENCODE GM12878 data but it should be explicitly stated in the figure legend.

We have partially answered this point in Point 4 of the Reviewer #1.

Following the Reviewer's suggestion, we have modified figure legends (Fig. 1d, Fig. 4b, Supp. Fig. 3c, Supp. Fig. 7b) to highlight the use of total counts and GM12878 cells.

4. Although the language in the article is generally acceptable, there are numerous minor grammatical errors that would easily be caught by a check from a native English speaker. One example:

P7 L194: Following comparable approach than for promoter usage ->

Following a comparable approach as for promoter usage

With the contribution from Dr. Richard Fish, now a co-author, we addressed grammatical issues over the entire manuscript.

Reviewer #3 (Remarks to the Author):

The authors have done a nice job with this analysis to identify regulatory variants associated with promoter usage and enhancer activity in EBV-transformed lymphoblastoid cells. The statistical analyses could be described better in some places, but overall appear to have been performed with high quality and the findings have been interpreted appropriately. One limitation is that conclusions are based predominantly on statistical analysis and associations identified from large-scale datasets, without the addition of definitive wet lab experiments to conclusively demonstrate regulatory activity at some of the identified loci. A possible concern is that the 154 samples do not appear to have been filtered for quality control which may raise questions concerning the influence of an outlying sample that is apparent in Supplementary Figure 2. Finally, some of the language was awkward and the manuscript could probably be improved by further editing. In my review I have provided several suggestions that the authors may consider in a potential revision of their manuscript in order to clarify certain aspects of the statistical and bioinformatic methods.

We thank the Reviewer for acknowledging the quality of our analysis and providing us with her/his comments regarding the limitations that we mention in more detail in the

revised manuscript.

(1) The study has investigated EBV-transformed lymphoblastoid cell lines. However, this cell type is not indicated in the title or the current Abstract. Since the effects of gene regulatory variants seem likely to be at least partially (if not strongly) cell type-specific, this would be important information to include in the Abstract and potentially the title as well.

We agree with the Reviewer comment and the choice of the cell model is now mentioned in the abstract.

Current version: *“Here we use natural genetic variation and CAGE transcriptomes from 154 EBV-transformed lymphoblastoid cell lines, derived from unrelated individuals, to map 5,376 and 110 regulatory variants associated with promoter-usage (puQTLs) and enhancer-activity (eaQTLs), respectively.”* (L.22-25).

(2) The authors do not appear to have applied quality filtering among the 154 samples and have included all 154 available samples in the analysis. This may be appropriate, although for example in Supplemental Figure 1 it is shown that sequencing depth varies 6-fold among samples. Does removal of samples with lower sequencing depth improve identification of puQTLs and eaQTLs associated with weakly expressed transcripts?

We thank the Reviewer for raising this point that we believe is due to lack of clarity in the Methods section of our manuscript.

We now mention that 8 samples were discarded for not reaching a minimum sequencing depth requirement.

Current version: *“Of the 164 samples originally sequenced, 8 with less than 5×10^6 mapped CAGE tags were discarded, in line with sequencing depth recommendations for the CAGE technology⁴⁷.”* (L.345-347).

(3) Likewise, in supplemental Figure 2, there is a sample shown in (A) and (B) that appears to be an outlier. Was the possibility of removing this sample considered? How sensitive are results to these potentially outlying value?

While we understand the Reviewer’s comment, we do not fully agree that an outlier appears on Supp. Fig. 2. While the potential outlier stands in the lower edges of PC2, it remains very much within other values of PC1. Still, we computed the *cis*-QTL mapping without the potential outlier and were able to replicate all the 5,376 reported puQTLs detected with the full dataset. Our puQTLs discovery analysis is not, therefore, sensitive to this particular sample.

(4) “We applied a minimum expression threshold on the mean expression over all individual included in the study of 0.5 RLE-TPM.” This procedure is reasonable, but it would be helpful to describe the justification of the 0.50 threshold. Could values above this threshold still be attributed to background noise? How does the value of 38,759 autosomal CAGE-peaks vary with changes in the RPE-TPM threshold applied?

We agree with the Reviewer that details about the justification of the expression threshold would be useful. We tend to adopt similar detection thresholds as previously used (*The FANTOM Consortium, Nature, 2014; Fort et al., Nature Genetics, 2014*).

We are now providing an additional panel on Supp. Fig. 1b that illustrates the variation of CAGE peaks passing different expression thresholds. The 0.5 threshold was chosen based on the curve, being the first point of the plateau following the exponential decrease, and no drastic decrease in CAGE peaks passing more stringent thresholds is observed.

(5) The study combines data from two different sets of samples, including the Coriell Cell Repository (CEU, n=86) and the GenCord collection (n=68). The samples do not appear to show systemic differences with regard to gene expression based upon principal component analyses, but was it possible to identify individual genes with significantly different expression between the two groups of samples?

The PCs projection shown on Supp. Fig. 2 is based on the CAGE peaks expression matrix following sample stratification correction with the first 3 PCs derived from genotype data and the first 20 PCs derived from promoter expression, as mentioned in the method section.

“We mapped puQTLs and eaQTLs using QTLtools (V1.1)¹⁷, with the following sets of covariates: for the puQTLs the first 3 PCs were derived from genotype data and the first 20 PCs were derived from promoter-normalized expression values. We controlled for stratification due to sample collections (Supplementary Fig. 2a) and library preparation batches (Supplementary Fig. 2b) in the normalized promoter usage data. For the eaQTL mapping, we used the first 3 PCs derived from genotype data and the first 12 PCs derived from enhancer-normalized activity values.” (L.379-385).

However before applying such corrections, stratification of CEU versus GenCord samples is observed on the first 2 PCs projection as expected (data not shown).

To gain in clarity we now mention in the figure legend that data have been residualized for the first 3 PCs derived from genotype data and the first 20 PCs derived from promoter expression values.

We determined the number of covariates used in the normalization of CAGE peaks quantification with a similar approach to that adopted by the Geuvadis consortium (Lappalainen et al., *Nature*, 2013). In brief, to correct for sample stratification and increase discovery power, *QTLtools* (Delaneau et al., *Nat Commun*, 2017) residualized phenotype data (here expression levels) for covariates using linear regression. It produces a phenotype matrix with quantifications that are independent of any of the covariates.

(6) Genotypes were imputed for data generated using the Illumina 2.5M Omni chip. Was there are particular r-squared value applied to ensure the imputed genotypes were of good quality?

We conducted the imputation procedure using IMPUTE2 (Howie et al., *PLoS Genet*, 2009), which provides an information score (*INFO*) strongly correlated with the *R-squared* value, as described in (Marchini J. & Howie B.N., *Nat Rev Genet*, 2010). The *INFO* metric takes values between 0 and 1, values near 1 indicate that a SNV has been imputed with high likelihood. However, there is no universal cutoff. Investigators have used cutoffs of 0.3 to 0.5 (Lappalainen et al., *Nature*, 2013; Waszak et al., *Cell*, 2015). In our study, we fixed the *INFO* cutoff at 0.5, as mentioned in the Methods section.

“The genotyping data for the CEU individuals included in our study were retrieved from the whole genome sequencing analyses performed by the 1000 Genomes Project Consortium¹⁴. We combined these two datasets and filtered for an information score above 0.5 and a minimum alternative allele counts of 10. We were left with genotype data at 7,508,202 autosomal variants for the 154 individuals.” (L.364-369).

(7) Lines 358 and 359. The authors state “No amplification bias was detected and individuals with suspicion of cross-sample contamination were removed from the study (Supplementary Fig. 1b).” It would be helpful to indicate here how many individuals were removed due to possible cross-sample contamination.

We agree with the Reviewer’s comment and have modified the “*Genotype-sequencing data consistency control*” method section accordingly.

Current version: *“Allelic consistency between genotype and CAGE tag sequences was assessed with the MBV methods (QTLtools package V1.1)¹⁶. Of the 156 samples passing the sequencing depth threshold, no amplification bias was detected and samples from 2 individuals, with suspicion of cross-sample contamination, were removed from the study (Supplementary Fig. 1c).”* (L.372-376).

(8) Lines 372 – 374. The authors state “To determine whole genome significance, corrected p-values for multiple variants within cis-windows were corrected for multiple

promoters or enhancers being tested, using the qvalue R package (V2.2.2)". It would be helpful for the authors to expand on this. Does this approach require that the p-values are independent?

We thank the Reviewer for raising this lack of clarity in our manuscript. We now provide more details about the statistical procedure developed in *QTLtools* (Delaneau et al., *Nat Commun*, 2017) to assess whole genome significance of observed association between variants and molecular phenotypes.

Current version: "*We delimited the set of variants to be tested per molecular phenotype by using topologically associating domains (TADs), as defined by Hi-C data on LCLs¹⁹. To determine whole genome significance, firstly 1,000 permutations were performed to adjust nominal p-values for the number of independent tests performed for each promoter or enhancer per cis-window. Secondly, adjusted p-values were corrected for the total number of promoters or enhancers tested genome-wide using the qvalue R package (V2.2.2)⁵⁴. We finally extracted puQTL or eaQTL with q-value<0.05 which corresponds to a 5% FDR.*" (L.386-393).

(9) It would be appropriate to include additional information regarding the 154 samples used in the analyses, in particular the sex of the samples. Was the number of males and females approximately balanced?

We agree with this comment and have added the sex of the samples on Supp. Table 1. Sex of the samples are balanced with 49% females (75/154) and 51% males (79/154).

(10) Supplemental Figure 3, part D. Are confidence intervals or standard errors shown for each transcription factor? It would be helpful to clarify this in the figure legend and possibly also indicate using a different colour which odds ratios are significant.

We modified the Supplementary Fig. 3d according to the Reviewer's suggestions. Odds ratio values passing the 5% FDR significance threshold are highlighted in red and the legend mentions that "bars show 95% confidence interval".

Reviewer #1 (Remarks to the Author):

The authors have addressed all of my concerns.

Reviewer #2 (Remarks to the Author):

The authors have addressed my concerns.

Reviewer #3 (Remarks to the Author):

I have reviewed the revised manuscript and the authors' response to my prior comments. I am satisfied that the authors' revision has been improved relative to the original submission and that they have addressed my comments with appropriate changes in the revised manuscript. I thus have no further comments at this time.